# Osteoporosis-Improving Effects of Extracellular Vesicles from Human Amniotic Membrane Stem Cells in Ovariectomized Rats

**DOI:** 10.3390/ijms26199503

**Published:** 2025-09-28

**Authors:** Ka Young Kim, Khan-Erdene Tsolmon, Zolzaya Bavuu, Chan Ho Noh, Hyun-Soo Kim, Heon-Sang Jeong, Dongsun Park, Soon-Cheol Hong, Yun-Bae Kim

**Affiliations:** 1College of Veterinary Medicine, Chungbuk National University, Cheongju 28644, Republic of Korea; 2Central Research Institute, Designed Cells Co., Ltd., Cheongju 28576, Republic of Korea; 3High-Technology Research Institute, ThanEver Inc., Daejeon 34054, Republic of Korea; hskim@thanever.com; 4Department of Food Science and Technology, Chungbuk National University, Cheongju 28644, Republic of Korea; 5College of Veterinary Medicine, Kangwon National University, Chuncheon 24341, Republic of Korea; 6Department of Obstetrics and Gynecology, Korea University College of Medicine, Seoul 02841, Republic of Korea

**Keywords:** osteoporosis, amniotic membrane stem cell, extracellular vesicle, growth factor, bone regeneration

## Abstract

Osteoporosis is a common skeletal disease characterized by decreased bone density, leading to bone fragility and fractures, especially in menopausal women. The purpose of this study is to confirm the anti-osteoporosis activity of stem cell extracellular vesicles (EVs) as a material of regenerative medicine. Mesenchymal stem cells have a potential to differentiate into osteocytes, so directly reconstruct bone tissue or facilitate bone regeneration via paracrine effects. Paracrine effects are mediated by functional molecules delivered in EVs released from stem cells. EVs containing high concentrations of growth factors (GFs) and neurotrophic factors (NFs) were attained via hypoxia culture of human amniotic membrane stem cells (AMSCs). From the EVs with a mean diameter of 77 nm, 751 proteins and 15 species of lipids were identified. Sprague-Dawley rats were ovariectomized, and eight weeks later, intravenously injected with EVs at doses of 1 × 10^8^, 3 × 10^8^ or 1 × 10^9^ particles/100 μL/body, weekly for eight weeks. One week after the final administration, the serum and bone parameters related to bone density were analyzed. Serum 17β-estradiol, alkaline phosphatase, and calcium levels that decreased in ovariectomized rats were restored by EVs in a dose-dependent manner. Bone parameters such as bone mineral density, bone mineral content, bone volume/tissue volume ratio, trabecular number, trabecular space, and bending strength were also improved by treatment with EVs. Such effects were confirmed by morphological findings of micro-computed tomography. Taken together, it is suggested that AMSC-EVs containing high concentrations of GFs and NFs preserve bone soundness by promoting bone regeneration and inhibiting bone resorption.

## 1. Introduction

Osteoporosis is one of the most common diseases nowadays. It is characterized by gradually decreased bone density and compromised bone strength, leading to bone fragility and fractures [1].

Osteoporosis is associated with postmenopausal estrogen deficiency [2], since the estrogen deficiency leads to an increase in bone remodeling, with an imbalance between bone formation and resorption [3,4]. Normal bone continuously requires bone remodeling in which osteoclasts and osteoblasts are coordinating well to regulate the remodeling process. The previous studies demonstrated that the bone remodeling process is damaged by estrogen deficiency due to the presence of estrogen receptors in osteoclasts. The bone resorption by osteoclasts increases, while the bone formation by osteoblasts decreases [5,6,7].

At present, bisphosphonates are commonly used as inhibitors of bone resorption, which is mostly in the osteoclasts’ metabolism for osteoporosis [8,9,10,11]. Bisphosphate compounds include alendronate, risedronate, iBandronate, zoledronate, etidronate, pamidronate, etc. Although widely prescribed, bisphosphonates have serious side effects such as nausea, gastric ulcer, and jaw osteonecrosis [9]. Therefore, it is necessary to develop a novel therapeutic without adverse effects, and cell-based therapeutic approaches should fulfill this requirement.

As a regenerative medicine for bone repair, stem cell therapies of osteoporosis have been increasingly attempted. Mesenchymal stem cells (MSCs) have the ability to differentiate into osteocytes, chondrocytes, and adipocytes [12]. Due to the potential of MSCs for osteocytic differentiation, they can be used for replacement therapy of destroyed bone tissue and osteoporosis. Otherwise, MSCs express and secrete numerous functional molecules including cytokines, chemokines, growth factors (GFs), and neurotrophic factors (NFs) which are involved in the paracrine effects of MSCs. Among them, GFs and NFs were found to be major mediators of tissue-regenerating paracrine effects. Indeed, MSCs release various GFs and NFs that are taken up by damaged tissues or cells. The GFs and NFs inhibit cell apoptosis as well as inflammatory tissue injury, facilitate cell proliferation, and thereby promote osteogenesis [13,14]. Notably, recent studies demonstrated that hypoxia-preconditioned MSCs intensify paracrine signaling [15,16].

Notably, it has been reported that functional molecules such as GFs and NFs are released in the form of extracellular vesicles (EVs) from functional cells, and exosomes are one of the EVs with a size of 50–300 nm [17]. EVs, as nano-lipid vesicles containing functional molecules, can easily penetrate bodily membrane barriers, so deliver the functional proteins to target cells. However, due to very low yield, low purity, and difficulty in isolation, EVs and exosomes have been limited in clinical application [18].

Recently, we attained large amounts of EVs via a hypoxic (2% O_2_) culture of human amniotic membrane mesenchymal stem cells (AMSCs). The EVs were confirmed to be 70–80 nm in diameter and contain high concentrations of GFs and NFs [19,20].

The present study aimed to verify the effects of EVs containing high concentrations of GFs and NFs, in comparison with zoledronic acid (ZA, a reference control), in promoting bone regeneration and in inhibiting bone resorption in ovariectomy (OVX)-induced osteoporosis conditions, and thereby to provide the possibility of the EVs as a candidate for osteoporosis treatment.

## 2. Results

### 2.1. Characteristics of AMSC-Derived EVs

Identification of EVs from normoxic (20% O_2_) and hypoxic (2% O_2_) culture media was confirmed via CD9, CD63, and CD81 markers. The CD-, CD63, and CD81 contents in hypoxia-cultivated EVs were higher than those from EVs obtained in a normoxic condition (Figure 1A).

In enzyme-linked immunosorbent assay (ELISA), the concentrations of insulin-like growth factor (IGF), fibroblast growth factor (FGF), transforming growth factor-β (TGF-β), vascular endothelial growth factor (VEGF), and platelet-derived growth factor (PDGF), that is, GFs and NFs related to osteogenesis, bone growth, angiogenesis, and tissue regeneration, were very low in normoxic culture medium. By comparison, it was confirmed that the GFs and NFs were very high in hypoxic culture EVs, reaching ten to a hundred times the normoxic culture EVs (Figure 1B).

In proteasome analysis, a total of 751 proteins were identified. Among them, the top 30 proteins are listed in Figure 1C and Appendix A. The top 10 proteins were identified as secreted protein acidic and rich in cysteine (SPARC), collagen α-1(I) chain, isoform 1 of fibronectin, filamin-A, collagen α-2(I) chain, vimentin, collagen α-3(VI) chain, collagen α-1(VI) chain, fibrillin, and collagen α-1(III) chain. SPARC is a critical matricellular protein that governs the proper assembly and maturation of collagen fibrils. Collagens and fibronectin are the fundamental building blocks for restoring tissue integrity following injury.

In lipidosome analysis, 15 lipid species were detected (Figure 1D) in which phosphatidylcholine (PC, 71.89%), ceramide (Cer, 14.31%), phosphatidylethanolamine (PE, 6.70%), and lysophosphatidylcholine (LPC, 2.37%) were the most abundant lipids, in comparison with relatively low levels of phosphatidylserine (PS), phosphatidylinositol (PI), and sphingomyelin (SM) (Appendix A). Such high concentrations of PC, Cer, PE, and LPC may give the EV membranes fluidity and stability.

From transmission electron microscopy (TEM), typical exosome structures of homogeneous, spherical, and membrane-bound vesicles were observed, in which the mean size of the EV particles was confirmed to be smaller than 100 nm (Figure 2A). NTA revealed the size distribution of EVs, wherein the major peak was found to be 77 nm (Figure 2B).

Notably, it was confirmed that the CM-DiI-labeled AMSC EVs readily entered both the normal and damaged (H_2_O_2_-treated) osteoblastic MC3T3-E1 cells, although a higher number of EVs was found in the injured cells (Figure 2C).

### 2.2. Osteoblast-Proliferative and -Protective Activities of EVs

Treatment of EVs (≥1 × 10^6^ particles/mL) significantly facilitated the proliferation of osteoblastic MC3T3-E1 cells in a concentration-dependent manner (Figure 3A). Furthermore, EVs significantly protected against 200 μM H_2_O_2_ (Figure 3B) and 2% hypoxic (Figure 3C) insults from the concentration of 1 × 10^6^ particles/mL.

### 2.3. Effects of EVs on Body Weights

The body weights of ovariectomized rats tended to increase, compared to Sham group animals (Figure 4). However, increased body weights were somewhat attenuated by treatment of the high-dose EVs (1 × 10^9^ particles/rat). Similarly, such body-weight-decreasing effect was also observed in ZA (100 μg/kg)-treated animals.

### 2.4. Effects of EVs on Organ Weights

After ovariectomy, in addition to body weights, liver and kidney weights tended to increase (Table 1). In contrast, spleen weight decreased following ovariectomy. Such changes in the weights of liver, kidneys, and spleen were somewhat recovered by treatment with EVs or ZA.

### 2.5. Effects of EVs on Serum Markers

The blood levels of bone-differentiation markers, including 17β-estradiol (Figure 5A), alkaline phosphatase (ALP; Figure 5B), and calcium (Figure 5C), decreased following ovariectomy, compared with Sham control. All the serum 17β-estradiol, ALP, and calcium were recovered by treatment with EVs, showing near-full recovery in the high-dose (1 × 10^9^ particles/body) group, although there were different responses among parameters. Such effects were also obtained by treatment with ZA (100 μg/kg).

### 2.6. Effects of EVs on Bone Minerals and Strength

Bone mineral density (BMD; Figure 6A) and bone mineral content (BMC; Figure 6B) significantly decreased in ovariectomized rats, compared to Sham control. However, the decreased BMD and BMC were restored by treatment with EVs, in which the high dose of EVs (1 × 10^9^ particles/body) markedly recovered both the BMD and BMC levels. Such BMD- and BMC-recovering effects were also achieved with ZA (100 μg/kg).

Ovariectomy induced significant decrease in bending strength of the femurs, in comparison with Sham control (Figure 6C). However, the reduced bone strength was recovered by treatment with EVs (1 × 10^9^ particles/body). Similarly, ZA also significantly recovered the bending strength.

### 2.7. Effect of EVs on Bone Structure

Three-dimensional images obtained by micro-computed tomography (micro-CT) showed severe loss of cancellous bones in ovariectomized rat femurs (Figure 7, ★), compared to normal features in Sham control. The cancellous bone loss was attenuated by treatment with EVs in a dose-dependent manner (1 × 10^8^–1 × 10^9^ particles/body). Especially, a marked recovery was found in the high-dose group (arrows). Such a cancellous bone-recovering effect was also seen in the femurs of rats treated with ZA (100 μg/kg).

### 2.8. Effects of EVs on Bone Structural Parameters

In order to quantitatively analyze the bone structure, the values of bone volume/tissue volume (BV/TV) ratio, trabecular thickness (Tb.Th), trabecular number (Tb.N), and trabecular space (Tb.Sp) of the femurs from the micro-CT images were measured (Figure 8). In comparison with the control bone (Sham), BV/TV, Tb.Th, and Tb.N markedly decreased in ovariectomized rat femurs (OVX). Notably, the bone structures were remarkably improved by EVs (1 × 10^9^ particles/body) or ZA (100 μg/kg).

The decreased BV/TV ratio in ovariectomized animals was significantly recovered by EVs (1 × 10^9^ particles/body) and ZA (100 μg/kg), although the effects were not statistically significant at lower doses of EVs (1 × 10^8^–3 × 10^9^ particles/body) (Figure 9A). By comparison, Tb.N was markedly restored by treatment with EVs (3 × 10^8^–1 × 10^9^ particles/body) and ZA (Figure 9C), while Tb.Th was not recovered by EVs or ZA (Figure 9B). Tb.Sp significantly increased following ovariectomy, which was attenuated by treatment with EVs in a dose-dependent manner (Figure 9D). Such Tb.Sp-recovering effect was also observed in the ZA-treated rat femurs.

## 3. Discussion

Osteoporosis is one of the most common symptoms in postmenopausal women. Recently, stem-cell-based therapy has been actively studied in tissue and bone regeneration [21,22,23]. In our recent study, it was confirmed that AMSC did not express MHC Class II-mediating immune reaction [24], so did not induce antibody formation. Importantly, EVs from AMSCs did not cause immunotoxicity from a safety test.

In the present study, we attained large amounts of EVs containing high concentrations of GFs and NFs from AMSCs via hypoxic culture (2% O_2_), wherein much higher functional molecules than normoxic culture (20% O_2_) were obtained. The EVs were confirmed with exosome-specific markers CD9, CD63, and CD81 (Figure 1) [19,20].

Proteomic analysis of AMSC EVs revealed a distinct protein signature characterized by a significant enrichment of proteins associated with the extracellular matrix (ECM) and tissue remodeling when compared to reference EVs (Appendix A). This profile suggests that the parent cells actively modulate their microenvironment through exosomal cargo, potentially promoting processes such as fibrosis, cell migration, and tissue repair. A key finding is the multiple structural ECM proteins and their modulators within AMSC EVs, especially high levels of collagen types. These proteins are the fundamental building blocks for restoring tissue integrity following injury. The delivery of these collagens via EVs may serve as a direct contribution to the nascent ECM at a damages site, accelerating the initial stages of tissue reconstruction [25].

This constructive process appears to be highly regulated, as evidenced by the dramatic abundance of SPARC. SPARC plays a critical role for proper assembly and maturation of collagen fibrils, ensuring the formation of a functional, organized matrix rather than disorganized scar tissue [26]. The high concentration of SPARC in AMSC EVs implies that they do not just supply raw materials, but also provides the molecular machinery to orchestrate effective tissue remodeling.

Furthermore, the pro-regenerative environment is actively protected, as indicated by the marked upregulation of tissue inhibitor of metalloproteinases 1 (TIMP1). During wound healing, a delicate balance between matrix synthesis and degradation by metalloproteinases (MMPs) is required. By delivering a high concentration of TIMP1, AMSC EVs can inhibit excessive MMP activity, thereby preserving the newly deposited ECM from premature breakdown and stabilizing the repairing tissue [27]. This triad of collagens (building blocks), SPARC (an organizer), and TIMP1 (a protector) represents a potent, coordinated system for constructive tissue repair.

Beyond directly remodeling the ECM, the cargo of AMSC EVs appears poised to modulate the behavior of recipient cells to support a regenerative cascade. The increased presence of filamin-A, a key cytoskeletal protein, is indicative of this. Exosomal filamin-A could enhance the migration and adhesion of resident cells, such as fibroblasts and endothelial cells, which is essential for wound closure and angiogenesis [28]. The detection of serotransferrin (TF), absent in reference EVs, further supports this hypothesis. Iron, transported by transferrin, is a vital cofactor for prolyl hydroxylase enzymes, which are indispensable for collagen synthesis [29]. By supplying transferrin, AMSC EVs may enhance the metabolic capacity of recipient cells to produce their own ECM, amplifying the regenerative effect. High contents of fibronectin and other ECM components might suggest that AMSC EVs promote a shift from a provisional fibronectin-rich matrix (typical of early wounds) to a more stable collagen-dominant matrix characteristic of mature and healed tissues [30].

In lipidome analysis, AMSC EVs showed a unique feature of lipid profile characterized by very high PC (71.89%) and Cer (14.31%), but low PI, PS, and SM compared with reference UCBSC EVs: in AMSC EVs and UCBSC EVs, 71.89% vs. 51.70% for PC, 14.31% vs. 1.52% for Cer, 0.00% vs. 10.36% for PI, 0.00% vs. 3.92% for PI, 0.32% vs. 21.344% for SM (Appendix A). Although any functional properties according to the compositional difference should be clarified, the high concentrations of PC, Cer, PE, and LPC may affect the fluidity and stability of EV membranes. Notably, the AMSC EVs did not show any alterations in the number, size, and conformational change such as aggregation.

Among many kinds of proteins, the concentrations of IGF, FGF, TGF-β, VEGF, and PDGF related to osteogenesis, bone growth, angiogenesis, and tissue regeneration, were very high in hypoxic culture EVs. In our previous reports, the GFs played key roles for the protection and regeneration of retinal ganglionic cells (RGC) and retinal tissue in glaucoma animals as well as keratinocytes and skin tissue in wound healing [19,20]. Notably, the size of AMSC EVs were 77 nm in diameter (Figure 2) which is much smaller than other stem cell EVs [31]. It was found that the CM-DiI-labeled AMSC EVs readily penetrated both the normal and damaged (H_2_O_2_-treated) MC3T3-E1 cells, as was also observed in retinal ganglionic cells (RGCs), keratinocytes, and fibroblasts [19,20]. Therefore, it is believed that the tiny AMSC EVs can more efficiently deliver functional molecules to the injured bone tissue.

Estrogen deficiency is one of the most representative characteristics of menopausal women. Since the limited bone restoration due to estrogen deficiency results from decreased osteoblast proliferation [5,6,7], we assessed the EVs’ activities on the MC3T3-E1 cell proliferation as well as protection against H_2_O_2_ and hypoxic insults. As expected from the high concentrations of GFs and NFs, EVs facilitated osteoblast proliferation and protected against oxidative and hypoxic stresses (Figure 3).

Since various functional proteins such as GFs and NFs have been demonstrated to control osteoclast activation, osteocyte differentiation, and bone formation [32], we treated ovariectomy-induced osteoporotic animals with various doses (1 × 10^8^–1 × 10^9^ particles/body) of EVs to confirm their beneficial effects.

It is well known that the lack of estrogen level in serum leads to increased body weight gain [33]. Also, the estrogen deficiency has been reported to damage some organs such as liver and kidneys [34,35,36]. It has been reported that MSC-derived exosomes could stimulate the repair of tissues such as liver, kidneys, and bones [37,38,39]. Our results were consistent with previous studies. Treatment with EVs gradually reduced the body weights increased by ovariectomy (Figure 4). In addition, the increased relative weights of the liver and kidneys in ovariectomized rats were restored after EV treatment (Table 1). Therefore, it is assumed that EVs not only reverse body weight gain, but also protect against liver and kidney damage following estrogen deficiency.

Serum markers are chosen to analyze metabolism related to bone resorption and formation. Previous studies investigated the decrease in serum level of estrogen after bilateral ovariectomy [40,41,42]. In the present study, there were decreases in 17β-estradiol, ALP, and calcium levels, to some extent, in ovariectomized rats (Figure 5). However, all three parameters were recovered by treatment with EVs or ZA, in spite of different sensitivities. Previous studies have reported that estrogen can also be synthesized by some tissues, other than ovaries, especially, adipose tissues [43,44,45]. Therefore, it is believed that EVs and ZA may stimulate alternative tissues to produce estrogen in ovariectomized rats, and thereby positively affect the bone restoration.

BMD is well known as a primary parameter for clinical diagnosis of osteoporosis. Some studies have reported amarked decrease in the values of BMD and BMC in eight weeks after ovariectomy [46,47]. In the present study, both the BMD and MBC significantly decreased eight weeks after ovariectomy. Therefore, we started treatment with EVs to the ovariectomized rats at eight weeks post-surgery. Most importantly, EV treatment improved bone mineral parameters to the levels of Sham control, which was also observed in ZA-treated animals (100 μg/kg) (Figure 6). In parallel with the decreased bone minerals, bending strength of the bone significantly decreased in ovariectomized rats. Notably, EVs remarkably restored the bone strength in ovariectomized rats, as seen in ZA-treated animals. The results indicate that EVs and ZA restore mechanical properties of the bones.

From the effects of EVs on BMD, BMC, and bending strength, we observed the inner integrity of femurs using micro-CT images. As expected, EV treatment inhibited bone loss in ovariectomized rats (Figure 7). This result can be considered to be similar to the effect of ZA. ZA is known to block bone resorption by inhibiting osteoclast proliferation and inducing the osteoclast apoptosis [48,49,50]. Although underlying mechanisms of EVs should be further clarified, such effects may be due to the activities of GFs and NFs in EVs, enhancing bone formation via osteoblasts, and inhibiting bone resorption via osteoclasts.

The bone-protective activities of EVs and ZA were confirmed by analyzing histomorphometric parameters, in which BV/TV can verify the mass of cancellous bone, while Tb.Th, Tb.N, and Tb.Sp represent the morphological structure of trabecular bones (Figure 8). Ovariectomy significantly decreased the BV/TV, Tb.Th, and Tb.N, and increased Tb.Sp (Figure 9). Notably, EVs and ZA increased Tb.N, that is, the number of bony trabecular, and thereby made the spaces between trabeculars (Tb.Sp) narrow, and enhanced BV/TV ratio.

## 4. Materials and Methods

### 4.1. Preparation of AMSCs

Human amniotic membrane tissues were obtained through cesarean section from a healthy pregnant female donor. The amniotic membrane tissues were digested with collagenase I, neutralized with an equal volume of medium containing 10% fetal bovine serum (FBS; Biowest, Kansas City, MO, USA), and centrifuged at 1500 rpm for 10 min. After washing twice, the contaminated red blood cells (RBCs) were lysed with RBC lysis buffer, and the remaining cells were suspended in Keratinocyte serum-free medium (SFM; Invitrogen, Carlsbad, CA, USA) supplemented with 5% FBS, 100 U/mL penicillin, and 100 mg/mL streptomycin (Invitrogen) [19,20]. Cultures were maintained under 5% CO_2_ at 37 °C in a culture flask. Media were changed every 2–3 days.

The prepared amniotic stem cells were analyzed for their stem cell markers in a flow cytometric system. The AMSCs were confirmed to be mesenchymal stem cells.

### 4.2. Preparation of EVs

The separated AMSCs were suspended in the defined serum-free medium in a Hyper flask (Nunc, Rochester, NY, USA) and cultivated under normal oxygen (20% O_2_, 5% CO_2_) or hypoxic oxygen (2% O_2_, 5% CO_2_) tensions at 37 °C for three days [19,20]. The media were filtered through a bottle-top vacuum filter system (0.22 μm, PES membrane) (Corning, Glendale, CA, USA). The conditioned media were 30-fold-concentrated using Vivaflow-200 (Sartorius, Hannover, Germany).

### 4.3. Western Blot Analysis of EV Markers

The isolated EVs were identified via CD9, CD63, and CD81 protein DC assay kit (Bio-Rad Laboratories, Hercules, CA, USA) [19,20]. An aliquot of normoxic and hypoxic culture EVs was denatured using denaturation buffer and then resolved via 12% SDS-PAGE. Resolved proteins were transferred onto Immobilon-P PVDF membrane and reacted with primary antibodies for CD9, CD63, or CD81 (1:1000; Abcam, Cambridge, UK) overnight at 4 °C, followed by incubation with horseradish peroxidase (HRP)-conjugated secondary anti-mouse antibody (1:2000; Abcam) at room temperature. After washing, the signal was recorded using WestFemto maximum sensitivity substrate kit under Bio-Rad ChemiDoc Imager (Bio-Rad Laboratories).

### 4.4. ELISA of GFs and NFs

ELISA was conducted to analyze functional molecules, that is, GFs and NFs, related to osteogenesis and tissue regeneration in normoxic and hypoxic culture EVs [19,20]. ELISA kits for IGF (ab211652; Abcam), FGF (ab219636; Abcam), TGF-β (ab100647; Abcam), VEGF (ab222510; Abcam), or PDGF (ab100622; Abcam) were used according to the manufacturer’s instructions. Briefly, EVs were put into the ELISA wells and incubated at room temperature. After washing 3–4 times, the primary antibodies were added, and reacted at room temperature. Following incubation with secondary antibody at room temperature, color-developing substrate was applied for 10–30 min. After treatment with a stop solution, the absorbance was measured at 450 nm.

### 4.5. Proteome Analysis of EVs

Proteome analysis was performed according to the standard protocol of KBio Health (Osong, Korea). For LC-MS/MS analysis of proteins, freeze-dried EV sample was reconstituted in 0.1% formic acid in water to a final concentration of 1 μg/μL. The protein concentration was determined using the BCA assay. Using a nano LC system, 3 μL (3 μg) EV sample was injected onto an LC analytical column (EASY-Spray 50 cm × 75 μm PepMap RSLC C18 2 μm) equipped with Q-Exactive^TM^ Plus Hybrid Quadrupole-Orbitrap^TM^ Mass Spectrometer (Thermo Fisher Scientific, Waltham, MA, USA), and separated with a linear gradient of solvent B (5–95%) over 70 min. The flow rate was maintained at 300 nL/min. Data analysis was performed in Proteome Discoverer 2.4 (Thermo Fisher Scientific), with searches run against the *Homo sapiens* database.

### 4.6. Lipidome Analysis of EVs

Lipidome analysis was performed according to the standard protocol of KBio Health. Using a UPLC system for analysis of lipids, 5 μL of sample was injected onto an LC analytical column (Agilent Poroshell EC-120 C18 2.1 × 100 mm) equipped with Q-Exactive^TM^ Plus Hybrid Quadrupole-Orbitrap^TM^ Mass Spectrometer, and separated with a linear gradient of solvent B (1–99%) over 30 min. The flow rate was maintained at 0.25 mL/min. Lipidomic database searching was performed using LipidSearch 4.2.21 (Thermo Fisher Scientific). Results were refined by removing peaks predicted to be false positives.

### 4.7. Transmission Electron Microscopy (TEM) of EVs

For TEM observation, EVs loaded on Formvar-coated copper grids were fixed with paraformaldehyde and glutaraldehyde, and counterstained with uranyl acetate to show the images of EVs. Using JEM-plus TEM instrument (Jeol, Tokyo, Japan) equipment with Oneview camera (Ga-tan, Berwyn, IL, USA), the images were attained at 30,000 magnification [20].

### 4.8. Nanoparticle-Tracking Analysis (NTA) of EVs

To analyze size distribution of EVs, NTA were performed using Nanosight NS 300 equipped with v3.2.16 analytical software (Malvern Panalytical, Malvern, UK). Purified EVs were adjusted to 10^8^ particles/mL (in accordance with the manufacturer’s recommendations) in PBS. The camera level was adjusted until the particle signal saturation did not exceed 20% and all particles were clearly visible, five images were captured in 60 s, and the size and particle concentration were analyzed [20].

### 4.9. Osteoblast Culture

Murine osteoblastic MC3T3-E1 Subclone 4 cells were purchased from American Type Culture Collection (ATCC CRL-2593, Manassas, VA, USA). Cells were cultivated in Dulbecco’s Modified Eagle’s Medium (DMEM; Biowest) supplemented with 10% FBS, 100 U/mL penicillin, and 100 mg/mL streptomycin [51,52]. The cultures were maintained under 5% CO_2_ at 37 °C in a culture flask. Media were changed every 2–3 days, and all experiments were conducted using the cells within the first five passages.

### 4.10. EV Uptake Assay

EVs in hypoxic culture medium obtained from AMSCs were labeled with red CM-DiI membrane dye (C7000; Invitrogen) and prepared at a concentration of 4 × 10^10^ particles/mL [19,20]. The prepared EVs were treated with CM-DiI membrane dye diluted (1:1000) with a stock solution prepared according to the manufacturer’s instructions, and incubated for 10 min at 4 °C. The labeled EVs were ultracentrifuged at 100,000× *g* for 4 h (Beckman, Brea, CA, USA), and the pellets were resuspended in phosphate-buffered saline (PBS) to make 50 μL/mL.

Cultivated MC3T3-E1 cells were seeded on 8-well chamber slides (NUNC C7182; Thermo Fisher Scientific, Waltham, MA, USA) at 1 × 10^5^/mL. After 24 h, cells were damaged with 200 µM H_2_O_2_ and then incubated with CM-DiI-labeled EVs (50 μL/mL) for 4 h at 37 °C [19,20]. The cells were fixed with 4% paraformaldehyde and treated with 0.1% Triton X-100 (Thermo Fisher Scientific). After blocking with 1% bovine serum albumin (BSA) for 1 h, the cells were immunostained with anti-α-tubulin antibody (1:1000, ab7291; Abcam) for 2 h at 37 °C, followed by goat anti-mouse IgG Alexa Fluor^TM^ 488 (1:500, Invitrogen) for 1 h at room temperature. The cell nuclei were stained with DAPI (Thermo Fisher Scientific) and examined under a microscope (BX51; Olympus, Tokyo, Japan) [19,20].

### 4.11. Osteoblast-Proliferative Activity

In order to assess the cell-proliferating activity of EVs, MC3T3-E1 cells (5 × 10^4^/mL) were seeded in a 96-well plate. The cells were treated with various concentrations (1 × 10^5^–1 × 10^7^ particles/mL) of EVs. After 24 h culture in a normoxic condition (20% O_2_) at 37 °C, the cell number was counted via an Evans blue-exclusion assay [19].

### 4.12. Osteoblast-Protective Activity

To evaluate the cytoprotective activity of EVs against oxidative stress, MC3T3-E1 cells (5 × 10^4^/mL) were seeded in a 96-well plate. The cells were exposed to 200 μM H_2_O_2_ and treated with EVs (1 × 10^5^–1 × 10^7^ particles/mL) [19]. Following 24 h culture in a normoxic condition (20% O_2_) at 37 °C, the cell number was counted.

To assess the cytoprotective activity of EVs against hypoxic injury, MC3T3-E1 cells (5 × 10^4^/mL) were seeded in a 96-well plate. The cells were treated with various concentrations (1 × 10^5^–1 × 10^7^ particles/mL) of EVs. After 24 h culture in normoxic (20% O_2_) or hypoxic (2% O_2_) conditions at 37 °C, the cell number was counted [19].

### 4.13. Design of Animal Experiment

Ten-week-old female Sprague-Dawley rats were purchased from DBL (Eumseong, Korea). All animal experiments were performed following protocols approved by the Institutional Animal Care and Use Committee (IACUC) of Chungbuk National University (CBNU), Korea (Approval No. CBNUA-1754-22-01; 23 June 2023). The animals were subjected to either bilateral ovariectomy or Sham operation without removing the ovaries described previously [53,54]. All rats were assigned to six groups at eight weeks post-surgery.

Ovariectomized rats were intravenously injected with EVs (1 × 10^8^, 3 × 10^8^ or 1 × 10^9^ particles/rat), ZA (100 μg/kg; a reference control), or their vehicle (PBS; 100 μL/rat) once a week for eight weeks.

Rats were sacrificed one week after the final EV administration. Serum and femurs from each rat were harvested, and then the serum and bone parameters related to bone remodeling were analyzed.

### 4.14. Analysis of Serum Markers

#### 4.14.1. ELISA of 17β-Estradiol

The level of 17β-estradiol in serum was measured using ELISA kits (Arigo Biolaboratories, Hsinchu, Taiwan) according to the manufacturer’s instructions.

#### 4.14.2. Biochemistry Analysis of ALP and Calcium

The levels of ALP and calcium in serum were analyzed using an automatic biochemistry analyzer (Hitachi 7180, Hitachi, Tokyo, Japan) according to the manufacturer’s instructions, respectively.

### 4.15. Analysis of Bone Parameters

#### 4.15.1. X-Ray Analysis of Bone Minerals

A dual-energy X-ray (PIXImus, Lunar, Madison, WI, USA) was used to measure bone mineral density (BMD; mg/cm^2^) and bone mineral content (BMC; mg) of the femurs. The rats were lightly anesthetized with diethyl ether, ventrally positioned on a DEXA table, and scanned according to the manufacturer’s procedures.

#### 4.15.2. Biomechanical Measurement of Three-Point Bending Strength

The bending strength of the femurs was measured via a three-point bending test. The test was carried out using Universal Material Testing Machine (Instron 4411, Instron, Norwood, MA, USA). The cross-head speed was set at 5 mm/min until fracture. The load–time curve was converted into a load–displacement curve, and then failure load was determined as a maximum bending strength at fracture in the load–displacement curve. The sample was set according to the literature [55].

#### 4.15.3. Micro-CT Analysis of Bone Structure

Micro-CT images of the femurs of rats were obtained using Quantum FX Micro-CT (Perkin-Elmer, Waltham, MA, USA), and reconstructed by scanner software (Quantum FX Micro-CT Control Software 4.1, Perkin-Elemer) [55,56]. Test conditions were set to a voltage of 90 kVp, current of 160 μA, and isotropic voxel size of 20 μm per pixel.

For the evaluation of bone regeneration, the histomorphometric parameters of bone were determined in the region of interest: i.e., BV/TV ratio (%), Tb.Th (mm), Tb.N (1/mm), and Tb.Sp (mm).

### 4.16. Statistical Analysis

The data were described as mean ± standard error. Statistical significance between the groups was analyzed by one-way analysis of variance (ANOVA) using the SPSS statistical software 30.0.0. (SPSS Inc., Chicago, IL, USA). *p*-values of less than 0.05 were considered statistically significant.

## 5. Conclusions

In the present study, we demonstrated that a high amount of EVs was obtained via a hypoxic culture of AMSCs. The AMSC EVs consist of 751 proteins and 15 lipid species including collagens, SPARC, TIMP1, PC, and Cer, which are distinct components related to tissue repair and EVs’ fluidity and stability. Especially, the EVs containing high concentrations of GFs and NFs improved ovariectomy-induced osteoporosis in rats. Actually, intravenous treatment of EVs increased blood estrogen level, ALP, BMD, BMC, and Tb.N, and thereby enhanced the bone bending strength. The results indicate that EVs can restore the bone soundness not only by directly inhibiting bone resorption and promoting bone regeneration, but also by indirectly enhancing estrogen and ALP.

### Limitations

Although we demonstrated AMSC EVs’ osteoporosis-improving potential, there are several limitations in the present study. EVs consist of many cellular components, but a part of parent cells, including lipids, proteins, and nucleic acids. Since limited information exists on the database of EV standards from different cells and collection conditions, we analyzed only AMSC EVs’ lipid and protein profiles, in comparison with reference UCBSC EVs. We could not show the fate of EVs in vivo, especially their penetration into bone tissue, due to a technical limitation of detection tools. Thus, it is not clear whether the bone-repairing effects of EVs result from direct (via GFs and NFs) or indirect (via estrogen recovery) activity. In addition, the effectiveness of EVs’ angiogenic activity, especially mediated by VEGF, on osteogenesis remains to be investigated.

## Figures and Tables

**Figure 1 ijms-26-09503-f001:**
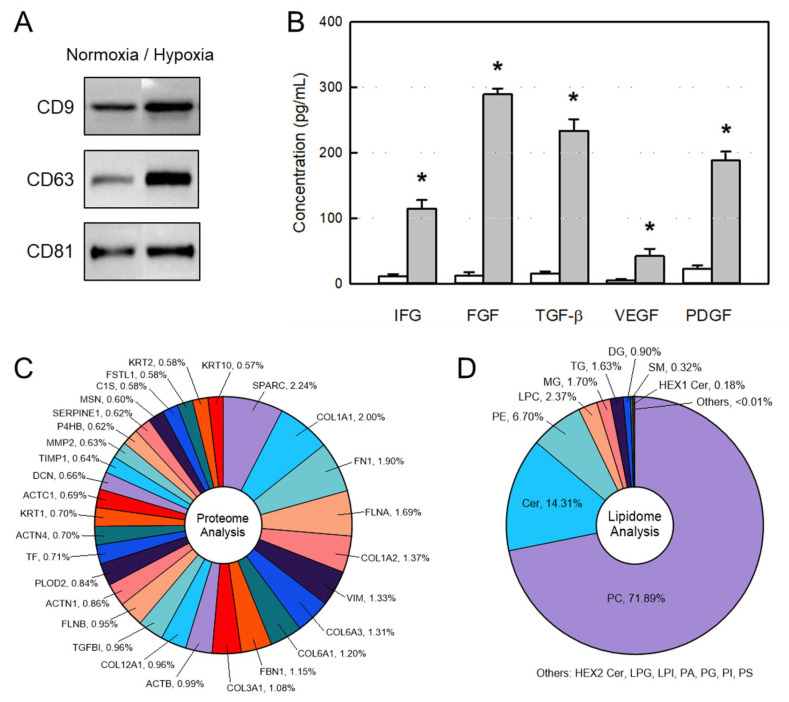
Components of extracellular vesicles (EVs) obtained from amniotic membrane stem cells (AMSCs). (**A**) Exosome markers CD9, CD63, and CD81 in normoxic and hypoxic culture EVs. (**B**) Concentrations of growth factors and neurotrophic factors in normoxic (white bars) and hypoxic (gray bars) culture EVs. (**C**) Contents of top 30 proteins identified from EVs. (**D**) Contents of lipids identified from EVs. IGF: insulin-like growth factor, FGF: fibroblast growth factor, TGF-β: transforming growth factor-β, VEGF: vascular epithelial growth factor, PDGF: platelet-derived growth factor. * Significantly different from each normoxic culture EV (*p* < 0.05).

**Figure 2 ijms-26-09503-f002:**
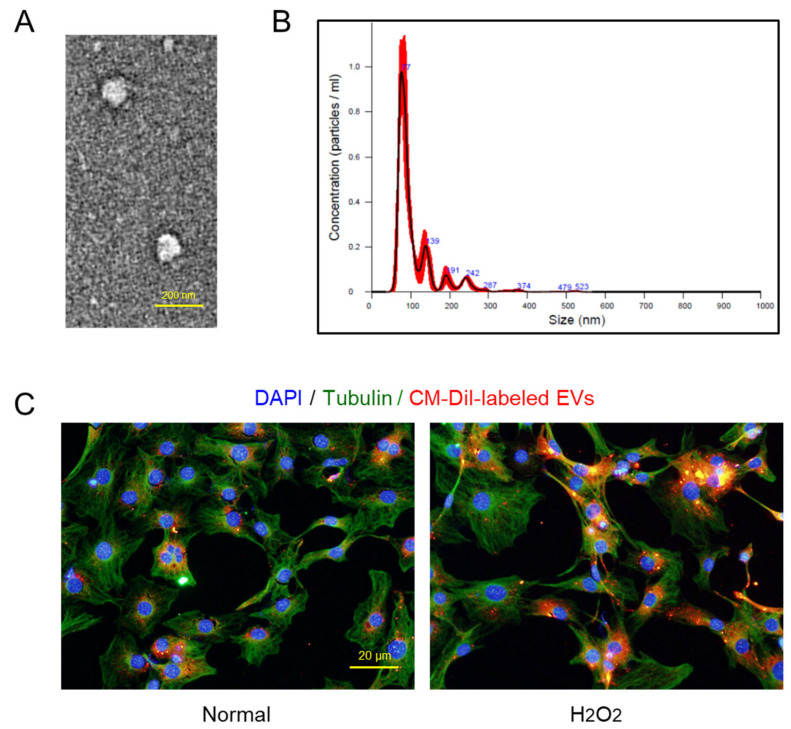
Size distribution and cell penetration of extracellular vesicles (EVs) obtained from amniotic membrane stem cells (AMSCs). (**A**) Representative transmission electron microscopic findings of EVs. (**B**) Particle size distribution of AMSC EVs analyzed by a nanoparticle-tracking analysis system. (**C**) Penetration of CM-DiI-labeled EVs into MC3T3-E1 cells exposed to 200 µM H_2_O_2_ or its vehicle (Normal). DAPI: 4′,6-diamidino-2-phenylindole.

**Figure 3 ijms-26-09503-f003:**
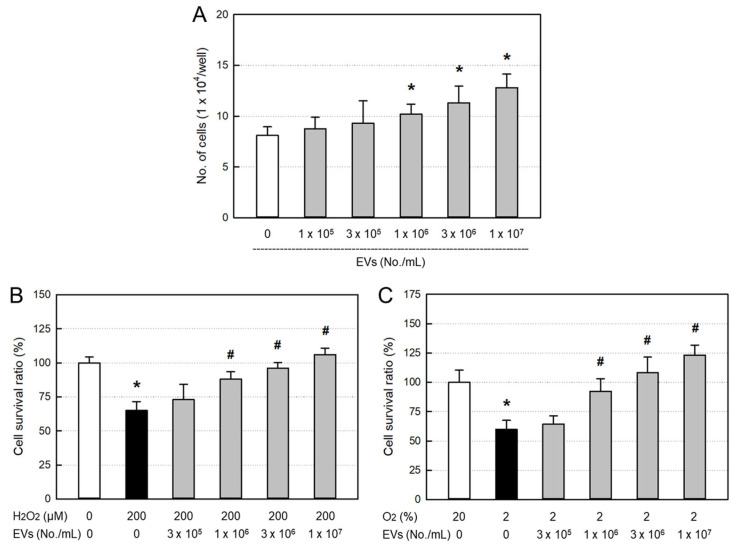
Osteoblast-proliferative and -protective activities of amniotic stem cell extracellular vesicles (EVs). (**A**) MC3T3-E1 cells were treated with EVs and incubated for 24 h. (**B**,**C**) MC3T3-E1 cells were treated with EVs and exposed to 200 μM H_2_O_2_ (**B**) or hypoxic (2% O_2_) condition (**C**) for 24 h. * Significantly different from normoxic (20% O_2_) control (*p* < 0.05). ^#^ Significantly different from H_2_O_2_ treatment or hypoxia (2% O_2_) alone (*p* < 0.05).

**Figure 4 ijms-26-09503-f004:**
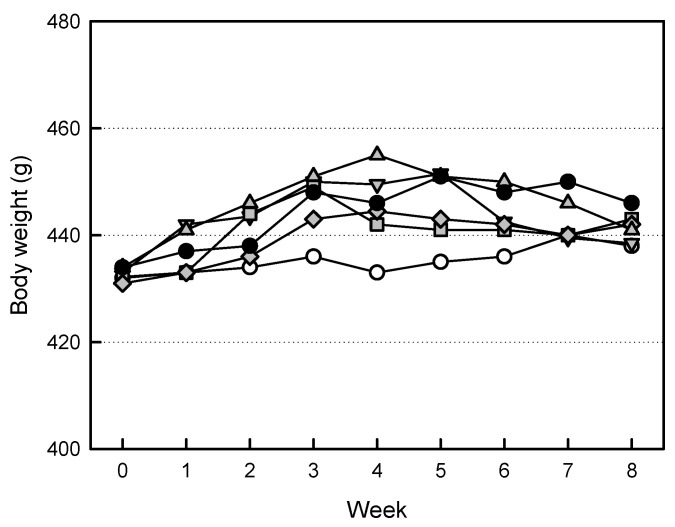
Change in body weight of ovariectomized (OVX) rats treated with extracellular vesicles (EVs) or zoledronic acid (ZA). ○: Sham control, ●: OVX alone, ▲: OVX + 1 × 10^8^ EVs/body, ▼: OVX + 3 × 10^8^ EVs/body, ◆: OVX + 1 × 10^9^ EVs/body, ■: OVX + ZA (100 μg/kg).

**Figure 5 ijms-26-09503-f005:**
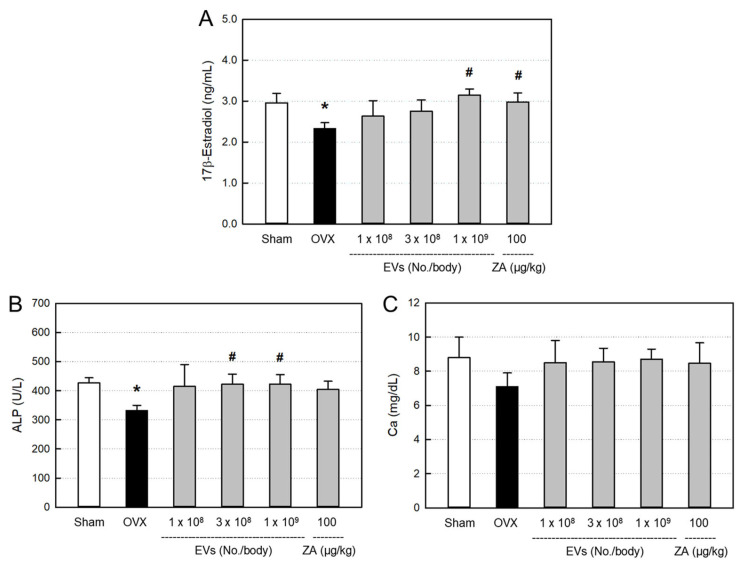
Blood 17β-estradiol (**A**), alkaline phosphatase (ALP) (**B**), and calcium (**C**) levels of ovariectomized (OVX) rats treated with extracellular vesicles (EVs; 1 × 10^8^–1 × 10^9^ particles/body) or zoledronic acid (ZA; 100 μg/kg). * Significantly different from Sham control (*p* < 0.05). ^#^ Significantly different from OVX alone (*p* < 0.05).

**Figure 6 ijms-26-09503-f006:**
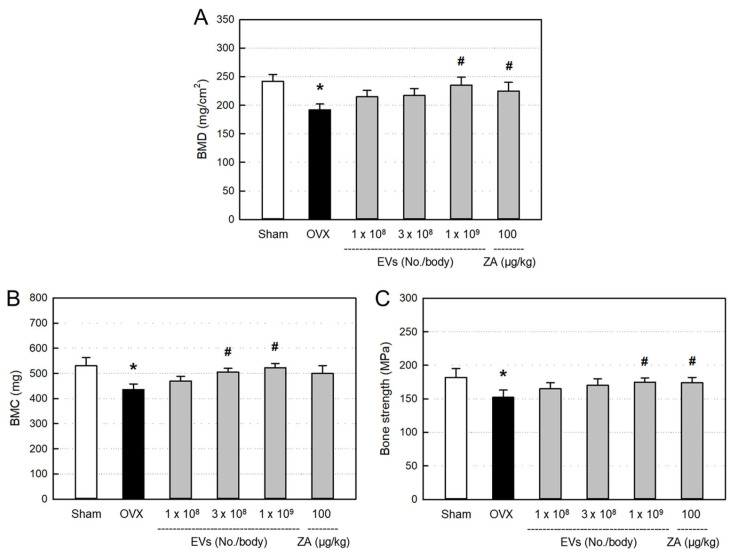
Bone mineral density (BMD) (**A**), bone mineral content (BMC) (**B**), and bending strength (**C**) of the femurs of ovariectomized (OVX) rats treated with extracellular vesicles (EVs; 1 × 10^8^–1 × 10^9^ particles/body) or zoledronic acid (ZA; 100 μg/kg). * Significantly different from Sham control (*p* < 0.05). ^#^ Significantly different from OVX alone (*p* < 0.05).

**Figure 7 ijms-26-09503-f007:**
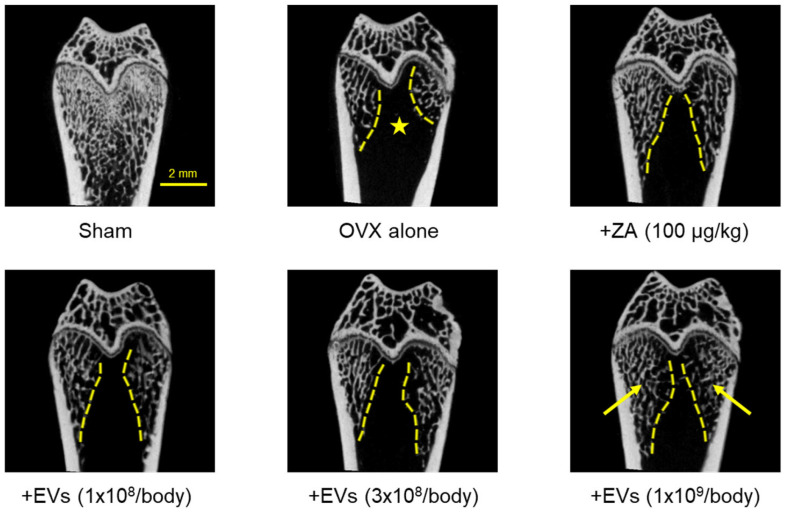
Representative 3D reconstruction of micro-computed tomographic images of the femurs of ovariectomized (OVX) rats treated with extracellular vesicles (EVs; 1 × 10^8^–1 × 10^9^ particles/body) or zoledronic acid (ZA; 100 μg/kg). The area of cancellous bone loss was marked with a star (★) and regenerated trabecular bones were marked with arrows.

**Figure 8 ijms-26-09503-f008:**
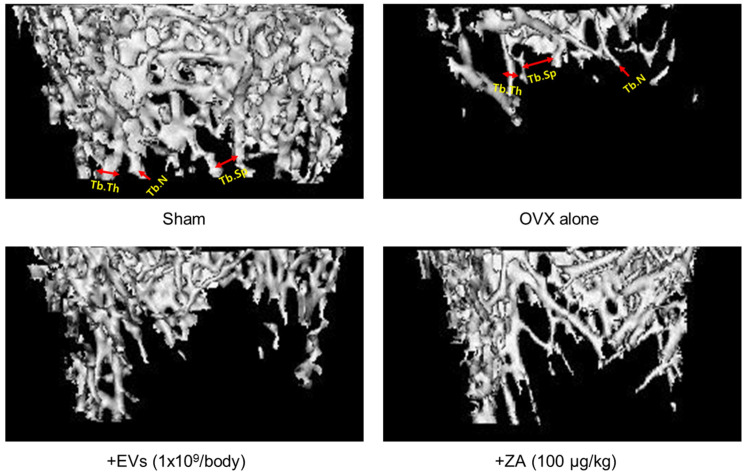
Representative histomorphometric parameters in the femurs of Sham control and ovariectomized (OVX) rats treated with extravesicles (EVs) or zoledronic acid (ZA). Tb.Th: trabecular thickness, Tb.N: trabecular number, Tb.Sp: trabecular space.

**Figure 9 ijms-26-09503-f009:**
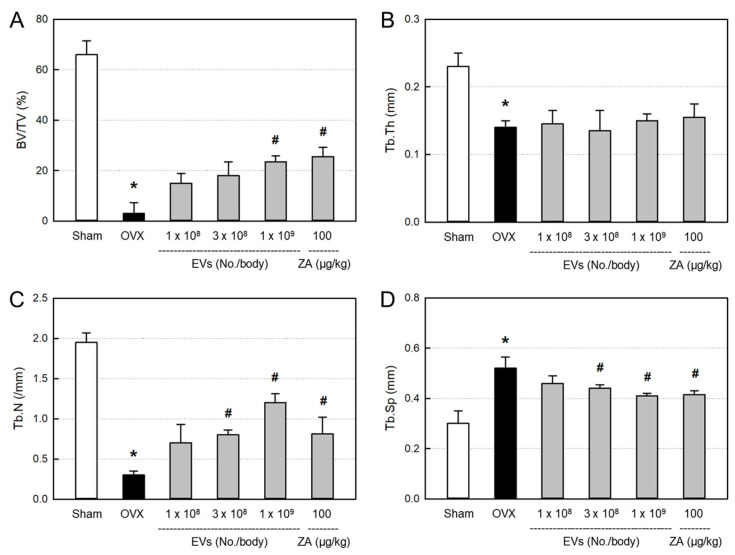
Histomorphometric parameters in the femurs of ovariectomized (OVX) rats treated with extracellular vesicles (EVs; 1 × 10^8^–1 × 10^9^ particles/rat) or zoledronic acid (ZA; 100 μg/kg). (**A**): bone volume/tissue volume (BV/TV) ratio, (**B**): trabecular thickness (Tb.Th), (**C**): trabecular number (Tb.N), (**D**): trabecular space (Tb.Sp). * Significantly different from Sham control (*p* < 0.05). ^#^ Significantly different from OVX alone (*p* < 0.05).

**Table 1 ijms-26-09503-t001:** The body and organ weights of ovariectomized (OVX) rats treated with extracellular vesicles (EVs) or zoledronic acid (ZA).

Weight		Sham Control	OVX Alone	+EVs (1 × 10^8^/Body)	+EVs (3 × 10^8^/Body)	+EVs (1 × 10^9^/Body)	+ZA (100 μg/kg)
Body weight (g)		434.3 ±37.7	442.9 ±30.9	439.7 ±30.0	446.4 ±18.0	435.5 ±20.4	441.7 ±17.6
Liver	Absolute (g)	8.99 ±0.21	9.68 ±0.94	9.63 ±0.27	9.62 ±1.01	9.48 ±0.80	9.36 ±1.25
	Relative (%)	2.06 ±0.19	2.19 ±0.09	2.19 ±0.13	2.16 ±0.20	2.15 ±0.14	2.15 ±0.26
Kidneys	Absolute (g)	1.82 ±0.12	2.04 ±0.22	1.86 ±0.08	1.86 ±0.20	1.90 ±0.26	1.82 ±0.64
	Relative (%)	0.42 ±0.01	0.46 ±0.02	0.42 ±0.02	0.42 ±0.04	0.43 ±0.06	0.42 ±0.18
Spleen	Absolute (g)	0.86 ±0.17	0.75 ±0.22	0.81 ±0.14	0.70 ±0.09	0.76 ±0.09	0.82 ±0.13
	Relative (%)	0.20 ±0.03	0.17 ±0.05	0.18 ±0.03	0.16 ±0.02	0.17 ±0.02	0.19 ±0.03

## Data Availability

Data are contained within the article.

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
