# Peer review of "Osteoporosis-Improving Effects of Extracellular Vesicles from Human Amniotic Membrane Stem Cells in Ovariectomized Rats"

_ijms, 2025, doi:10.3390/ijms26199503_

Round 1
Reviewer 1 Report
Comments and Suggestions for Authors
The manuscript explores, the efficacy of conditioned AMSC-derived EVs as therapy in OVX rat model in reference to ZA. Since EVs are reported to have positive effect in bone regeneration, the current study addresses an important problem. It demonstrates potentially impactful work but needs to be modified to enhance scientific integrity as well as translational relevance. The following issues need to be addressed:
- The EV isolation method is not is only based on filtration and not on SEC or UC or gradient based protocol which might lead co isolation of soluble proteins or lipoprotein. The authors are encouraged to provide documentation of absence of protein aggregates. Also, the characterization is not robust. The particle size distribution needs to be analysed using NTA, and also TEM data is required. Provide cellular uptake results of EVs also. The characterizations should be presented following the MISEV2023 guidelines. The marker panel (western blot) needs to be expanded according to MISEV2023 as it lacks negative control
- Also, the text alternatively mentions exosomes and EVs. The authors should be consistent in the nomenclature terms after validating through NTA analysis as these are based on size.
- The authors should consider providing particle to protein ratio.
- The authors should confirm the ELISA report is due to EVs only and not co-concentrated soluble proteins and thus should include in supplementary file experimental data of non-EV fractions/EV depleted media at the same protein to particle dose.
- For osteoblast proliferation what assay method did the authors use? They are also encouraged to include osteoblast differentiation markers (RUNX2/OCN etc or alizarin red etc.) in addition to proliferation markers.
- Biodistribution of EVs: It is well known in literature that after i.v. administration the EVs tend to accumulate in liver, spleen etc with variable amount in bone marrow (unless primed/injured). Thus, the authors are requested to provide whole body imaging (ivis/ radiolabeled) with ROIs for bone uptake quantification.
- In serum markers, in addition to ALP, serum PINP and betaCTX-1 (bone turn over markers) need to be incorporated to identify bone formation and resorption respectively.
- In order to supplement biocompatibility serum marker panels (AST/ALT, creatinine, BUN) and detailed histo-morphological evidence if all major organs need to provided.
- For the in vivo data presentation please provide per animal dot plots with mean+/- SEM/SD and state exact n’s. In micro-CT data add scale bars, consistent units.
- It would be exciting to observe, whether the effect persist with proteinase treated EVs or after RNAse/DNAse treatment to figure out active cargo components and also profile EV proteome/ miRNome.
- Without absolute comparative analysis, the authors should refrain from boldly claiming EVs superiors to ZA at higher doses.
Reviewer 2 Report
Comments and Suggestions for Authors
The manuscript investigates the therapeutic potential of extracellular vesicles (EVs) derived from hypoxia-preconditioned human amniotic membrane stem cells (AMSCs) in ovariectomized rat models of osteoporosis. The authors demonstrate that AMSC-EVs enriched in growth and neurotrophic factors improve osteoblast proliferation, restore serum markers, and enhance bone density and microarchitecture compared with control and zoledronic acid. The topic is timely, and the data are of potential translational significance.
Questions:
- How reproducible are the EV preparations across different AMSC donors?
- Did the authors confirm the particle size distribution (e.g., via NTA or DLS) in addition to Western blot markers?
- Can the authors provide evidence that estrogen restoration is directly mediated by EV cargo?
- Were bone turnover markers (e.g., osteocalcin, CTX) measured in addition to estradiol, ALP, and calcium?
- Were histological assessments of osteoblast/osteoclast activity performed to support micro-CT findings?
- How do the authors address potential off-target or systemic effects of repeated IV EV injections?
- Did the authors assess potential immune responses against human-derived EVs in rat recipients?
- What is the rationale for the selected dosing regimen (weekly for 8 weeks)? Would more frequent administration enhance outcomes?
- How stable are the EVs during storage? Was potency confirmed after thawing?
- Could these EVs be scaled up under GMP conditions for potential clinical translation?
- The manuscript sometimes uses “exosomes” and “EVs” interchangeably. Strictly speaking, the isolated vesicles are EVs; the term “exosomes” should be avoided unless biogenesis is demonstrated.
- Some micro-CT images (Fig. 6) appear low resolution. Annotated arrows/stars could be clearer.
- The potential immunological effects of EVs (immunogenicity, biodistribution) are not discussed and should be acknowledged as limitations.
Reviewer 3 Report
Comments and Suggestions for Authors
The authors aimed to verify the effects of EVs containing high concentrations of GFs and NFs, in comparison with zoledronic acid.
In the abstract, the authors should include the aim of the experiment.
Few numerical values of results should be included in the abstract.
A maximum of five keywords from MeSH may be enough.
Ethical approval number and other details should be added.
Line 354- what is the reason behind the time period of 8 weeks?
In the manuscript, the authors should add more facts on the size of extracellular vesicles. What can be the effect on vascular endothelial cell proliferation? What happens to the blood supply to the bone?
The authors should clearly state in the paper what is unknown in the existing literature and what new scientific facts are added through the present study.
What can be major challenges using EVs? How to standardize?
What are the steps to tackle degradation?
Limitations of the study were missing.
The conclusion section is very weak. This section should be re-written with implications of the results rather than highlighting the results. The content of the text should be elaborate rather than being too short.
Round 2
Reviewer 1 Report
Comments and Suggestions for Authors
The study can be designed and executed better